# Design and Shear Analysis of an Angled Morphing Wing Skin Module

**Jinrui Yu and Jiayao Ma ***

School of Mechanical Engineering, Tianjin University, 135 Yaguan Road, Tianjin 300350, China; yjr@tju.edu.cn
* Correspondence: jiayao.ma@tju.edu.cn

**Featured Application: The research focuses on the design of a shear morphing wing skin, which can potentially greatly expand the shear morphing capability of aircraft wings and meet the requirements of a large swept-back angle at high speed and a small swept-back angle at low speed during flight.**

**Abstract:** Morphing wing skin can greatly improve the performance of aircraft by adjusting the shape of the wings according to different flight conditions. However, it is a challenge to maintain a smooth aerodynamic wing skin surface during the deformation process. Here, we propose an angled morphing wing skin module based on a silicon rubber matrix reinforced by carbon-fiber-reinforced polymer rods, which takes advantage of the tensile stress generated during shear to prevent it from wrinkling under large shear deformation. Experiments conducted on a series of wing skin modules with varying initial angles indicate that by starting from an angled configuration, the skin module can withstand a pure shear deformation of 92° without wrinkling, 53% larger than existing designs. A parametric analysis was also conducted to analyze the effects of geometric and material parameters on the wrinkle-free deformation range. Finally, a theoretical model based on the energy method was developed to unveil the underlying wrinkle prevention mechanism and to estimate the critical wrinkling angle of the skin. The proposed design can potentially greatly expand the shear morphing capability of aircraft wings, leading to larger variation in sweepback angle and therefore superior aerodynamic performance.

**Keywords:** aircraft wing; morphing wing; shear morphing; composite skin; wrinkle control

---

## 1. Introduction

Morphing aircraft can change the shape of their wings to adapt to different flight conditions of speed, altitude, and angle of attack [1]. Past research has shown that a small sweep angle can increase the lift coefficient for takeoff and landing at a low speed, while a large sweep angle can reduce high-speed drag and is suitable for high-speed cruising [1]. Large span wings can increase lift while small ones can reduce parasitic drag [2]. Morphing aircrafts greatly improve the versatility and efficiency of the entire flight process of takeoff, climb, cruise, descent, and landing [3,4] and are considered a promising technology for aircraft in the future [5]. Morphing skins are the main component that forms the aerodynamic profile of morphing aircrafts [6]. However, the skin has been a bottleneck restricting the development of morphing aircrafts. This is because the skins need to bear aerodynamic loads in the out-of-plane direction, undergo large deformation in the actuating direction, and maintain a smooth surface during deformation [7]. Therefore, they are required to meet the contradictory requirements of high out-of-plane stiffness and low in-plane stiffness [8–10], which results in great challenges to the design of the skins.

Existing morphing skins can be divided into the following five categories: segmented skins, honeycomb skins, corrugated skins, shape memory polymer (SMP) skins, and rubber skins. Segmented skins [11,12] are deformed by relative sliding between discrete scales.

---

This kind of design provides large deformation and sufficient out-of-plane stiffness. However, the surface is not smooth and continuous, and thus not able to maintain a seamless aerodynamic surface for the aircraft [6]. Honeycomb structures supported by a membrane to form a seamless surface have been studied for the skin design [13,14]. Honeycomb structures are able to maintain a high out-of-plane stiffness to carry aerodynamic pressure while the in-plane stiffness is low to reduce actuation [15]. However, limited by the elastic limit of the base material (metal or plastic), they can only achieve a small range of stretch, torsion, and sweep deformation [16,17]. The corrugated structures are considered a good candidate for morphing applications due to the high out-of-plane stiffness and in-plane transverse compliance [18]. Yokozeki et al. [19]. proposed a corrugated wing skin reinforced by unidirectional carbon-fiber-reinforced polymer rods, but only for bend and stretch deformation, rather than sweep deformation. As a result of the development of smart materials, shape memory polymers (SMPs) are increasingly applied to the design of the skins. SMPs have the characteristics of large deformation and adjustable stiffness under external excitation [20,21], and are widely applied in the research of sweep, torsion, and bend deformation [22–24]. Nevertheless, many studies have shown that dynamic control problems, including uneven temperature distribution during heating [25], peeling of the hot wire and shape memory polymer [26], and cracking of the SMPs after multiple cycles of heating and deformation [27], have not been overcome. Finally, rubber can realize various forms of deformation [28,29] due to its high elasticity, but it tends to wrinkle during the deformation process. To address this issue, Asheghian et al. [30]. achieved a 45° shear deformation without wrinkling by applying pre-straining to the rubber, but this posed the problem of creep. Wu R et al. [31]. designed a kind of rubber matrix composite skin reinforced by carbon-fiber-reinforced polymer (CFRP) rods and Kevlar-fiber-reinforced polymer (KFRP) without pre-straining, and achieved shear deformation over 60°. However, wrinkling is still prone to occur at a large shear angle, thus limiting the deformability of the morphing wing skin.

In this paper, we propose a new angled wing skin based on reinforced rubber to achieve smooth and large shear deformation. By designing the initial angle of the skin, the tensile stress generated during a shear process is used to delay the occurrence of wrinkling and thus expand the deformation range. The layout of the paper is organized as follows. In Section 2, the geometry of the skin and the fabrication process are introduced. Then the experiment and finite element modeling are presented in Section 3. The deformation mechanism of the skin and the effect of design parameters are discussed in Section 4. A theoretical model is derived to explain the wrinkle prevention mechanism in Section 5. Finally, a summary conclusion is given in Section 6.

## 2. Materials

### 2.1. Design

The skin module consists of a rubber matrix, one layer of CFRP rods, two layers of KFRP fibers, and a frame, as shown in Figure 1. The CFRP rods are parallel to the edge of the skin, and the KFRP fibers are laid parallel to each other on the top and bottom sides of the CFRP rods, and then embedded in the silicon rubber matrix to form a smooth outer surface. Moreover, a frame composed of strips and pins was designed to fix the rubber matrix. On each strip, several through-holes are made to mount the pins. The pins passed by the CFRP rods are devised to be rotatable, thus acting as a connection between the CFRP rods and the frame. The module can be parameterized by the global length $L$, width $W$, thickness $H$, the initial angle $\theta_0$, the distance $l$ between the CFRP rods and the top surface of the rubber matrix, the intervals $l_1$ and $l_2$ of the CFRP rods and KFRP fibers, and the diameter $d_1$ and $d_2$ of the CFRP rods and KFRP fibers.

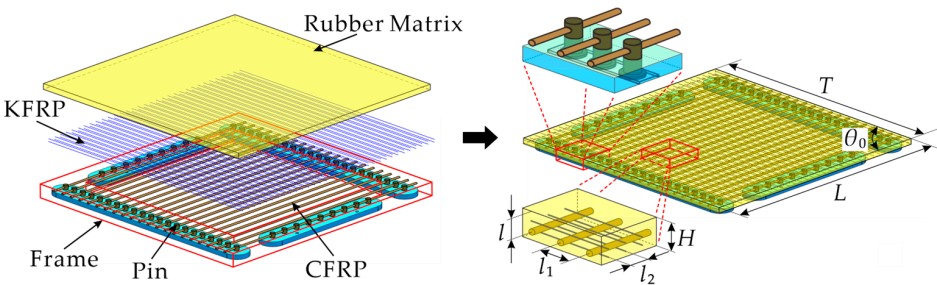

**Figure 1.** Design of the morphing wing skin module.

### 2.2. Fabrication

Silicon rubber (MVQ-4012), KFRP fibers (1414), and CFRP rods (T300) were adopted to fabricate the skin through a four-step procedure illustrated in Figure 2. First, the CFRP rods were passed through the pins on the frame. It should be noted that two manganese steel rods (diameter = 1.5 mm, modulus = 265 GPa) were installed on the two sides instead of CFRP rods to strengthen the boundaries where overstretching of the rubber matrix is likely to happen. The CFRP skeleton is shown in Figure 2a. Second, a mold with pillars on two sides was fabricated. The KFRP fibers were wound around the pillars and laid evenly on the top and bottom sides of the CFRP rods. The initial angle was determined by the shaping plate, as shown in Figure 2b. Third, silicon rubber was stirred fully and injected into the mold, as shown in Figure 2c. Finally, the silicon rubber was cured at room temperature for 12 h, and then the specimen shown in Figure 2d was obtained by removing the mold and pruning. Five specimens with different initial angles $\theta_0 = 50°, 60°, 70°, 80°, 90°$ were manufactured and named M50, M60, M70, M80, M90, respectively. The other geometric parameters are listed in Table 1. Note that the distance $l$ was chosen as 4.5 mm, indicating that the CFRP rods were closer to the bottom surface of the rubber matrix. This was because an offset of the CFRP rods led to a smoother top surface (detailed experimental results in Appendix A).

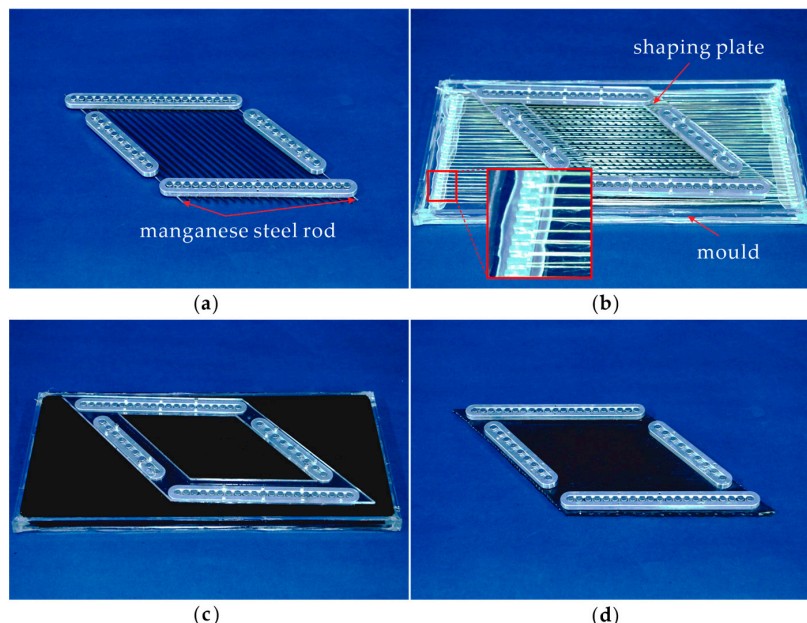

**Figure 2.** Fabrication procedure of the morphing wing skin specimen: (**a**) skeleton made of CFRP rods, manganese steel rods, and the frame; (**b**) mounting KFRP and shaping the initial angle of the specimen; (**c**) injecting rubber and curing; (**d**) removing the mold, pruning and obtaining the specimen.

**Table 1.** Design parameters of the specimens.

| Parameters | Values |
| --- | --- |
| Length $L$ | 140 mm |
| Width $W$ | 140 mm |
| Thickness $H$ | 6 mm |
| Distance $l$ | 4.5 mm |
| Interval of CFRP rods $l_1$ | 7 mm |
| Interval of FKRP fibers $l_2$ | 4 mm |
| Diameter of CFRP rods $d_1$ | 1.5 mm |
| Diameter of FKRP fibers $d_2$ | 0.05 mm (400 D) |
| Initial angle $\theta_0$ | $50°, 60°, 70°, 80°, 90°$ |

## 3. Methods

### 3.1. Experiment

We performed shearing experiments of the angled specimens on an Instron universal testing machine (type 5982). A planar four-bar linkage, as shown in Figure 3a, was designed to mount the specimen with bolts, which was then attached to the testing machine through a crosshead chuck. To eliminate dynamic effects, the loading rate was 10 mm/min. Considering the Mullins effect, i.e., the stress softening phenomenon after prestressing [32], each specimen was sheared three times before the formal experiment. All specimens were loaded until wrinkling occurred. As shown in Figure 3b, the precise deformed configuration was obtained by a 3D scanner (Artec Space Spider, Artec 3D, Luxembourg) with a scanning accuracy of 0.05 mm at a machine temperature of 37 °C. The configuration was scanned and recorded every 2°. Moreover, to quantify the deformation process of the skin, the strain field of the specimen was captured by a digital image correlation (DIC) system CSI Vic-3D9M at a frame time range of 500 ms, as seen in Figure 3c, and speckles with a density of about $1.4/\text{mm}^2$ were sprinkled on the surface of specimens. The deformation processes of specimen M90 and specimen M50 are shown in Figures 4a and 4b, respectively.

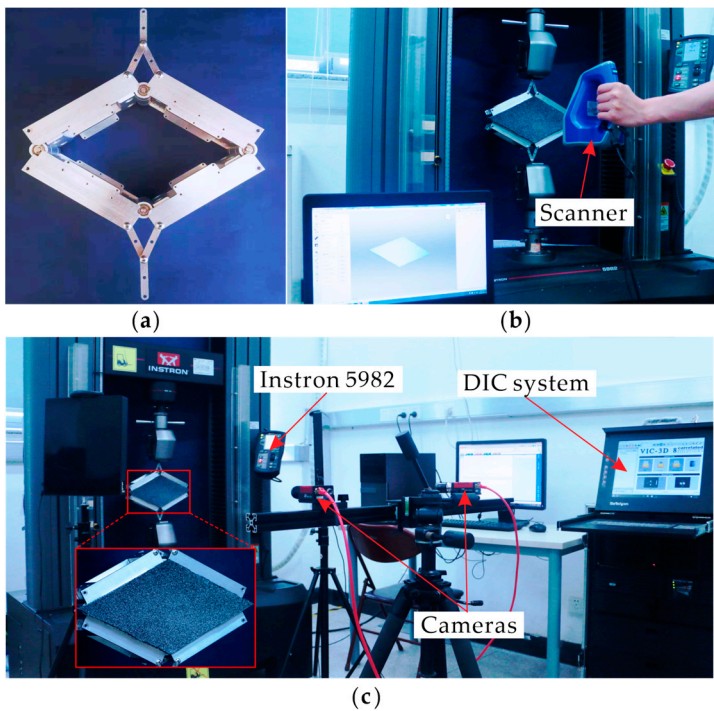

**Figure 3.** Experimental setup: (**a**) planar four-bar linkage; (**b**) 3D scanner for precise deformed configuration; (**c**) DIC system for strain field of the specimen.

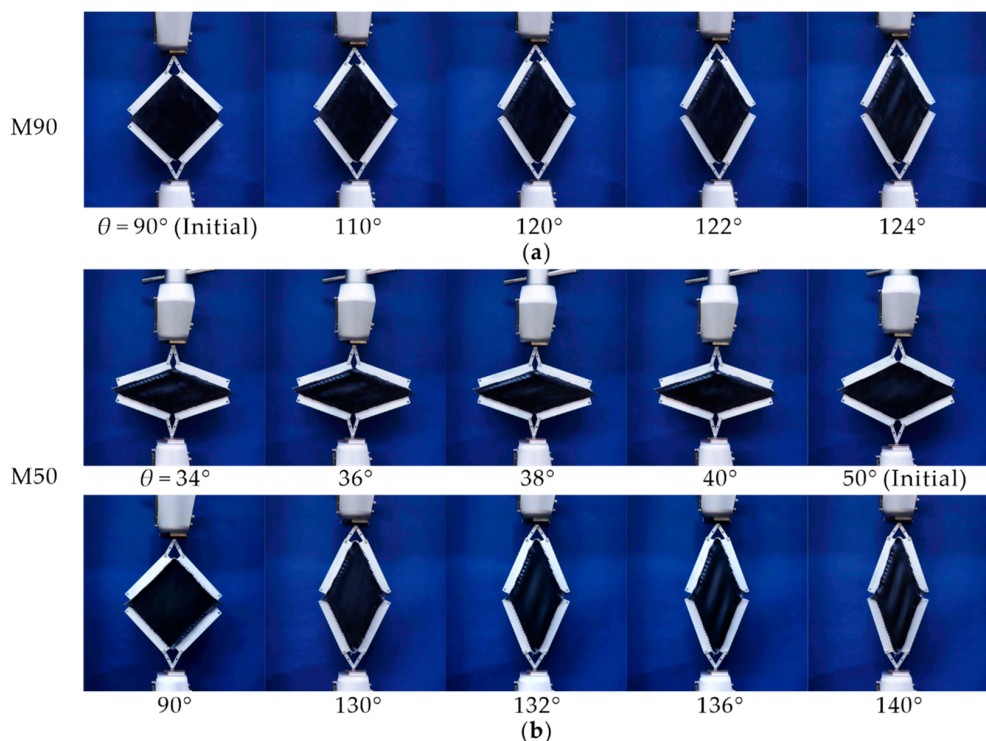

$\theta = 90°$ (Initial)     110°     120°     122°     124°

(**a**)

$\theta = 34°$     36°     38°     40°     50° (Initial)

90°     130°     132°     136°     140°

(**b**)

**Figure 4.** Deformation processes of the specimens: (**a**) M90; (**b**) M50.

### 3.2. Finite Element Modeling

To understand the shearing deformation mode and critical wrinkling angles of the skin in more detail, numerical models were established in Abaqus/Riks. The rubber matrix was modeled with a hexahedral solid element C3D8RH to simulate its hyperelasticity and incompressibility. Beam elements, B31, and Truss elements, T3D2, were used to mesh the CFRP rods and the KFRP fibers, respectively. The mesh size of 1 mm was found to be appropriate for the three components and adopted after conducting a careful mesh convergency test. The Yeoh model was selected as the constitutive model for rubber since it was appropriate to simulate the large deformation behavior of rubber [33,34], whereas a linear elastic model was adopted for the CFRP rods and KFRP fibers. The material properties of the three components of the skin module are listed in Tables 2 and 3, respectively. Moreover, since the shear wrinkling was sensitive to geometric imperfection, the first five buckling modes for each model were obtained through a linear buckling analysis and introduced into the model as initial defects.

**Table 2.** Material properties of CFRP and KFRP.

| Material | $\rho$ (g/cm$^3$) | $E$ (GPa) | $\mu$ |
|---|---|---|---|
| CFRP | 1.8 | 210 | 0.3 |
| KFRP | 1.4 | 104 | 0.1 |

**Table 3.** Material property of rubber.

| $\rho$ (g/cm$^3$) | $C_{10}$ | $C_{20}$ | $C_{30}$ | $D$ |
|---|---|---|---|---|
| 1.8 | 0.05905 | $-0.006268$ | 0.0010344 | 0 |

Four rigid links connected by hinges were used to generate the shear deformation as in the experiments. Link-1 was fixed while a prescribed displacement was assigned to link-2 to control the shear angle. To resemble the actual boundary conditions, a tie connection was

applied between the module and link-1 and link-2, and a hinge connection was adopted between the module and the other two links. The friction of the hinge between the rubber and the links was set as 0.8 through multiple attempts and comparisons with experiments. An embedded connection was set up to perform the interaction between KFRP fibers and CFRP rods and the rubber matrix. The finite element model is shown in Figure 5.

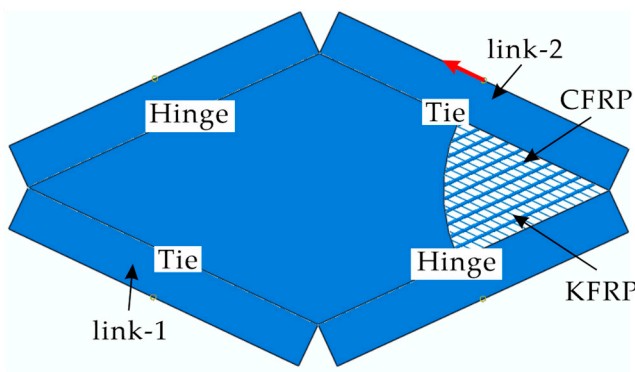

**Figure 5.** Finite element model of the skin module.

*3.3. Validation*

The numerical results of specimen M50 were compared with the experimental results to validate the numerical model. As shown in Figure 6a, four configurations near the critical wrinkling state of the skin with θ of 130°, 132°, 136°, and 140°, from which it can be found that the same wrinkling mode was obtained with the experiment. Moreover, the differences in U3, which is the displacement of the skin module in the z-direction, of the numerical and experimental configurations at the four morphing angles, were calculated and normalized by the thickness of the module. The pie graphs of the errors shown in Figure 6c also indicate that the numerical and experimental results are very close. The principal stresses at the intersection of the diagonal lines of the module when the skin is in the ideal wrinkling-free state are calculated as follows based on the Yeoh hyperelastic constitutive model [33]:

$$\sigma_i = \frac{\partial W}{\partial S} = \left( C_{10} + 2C_{20}(I_1 - 3) + 3C_{30}(I_1 - 3)^2 \right)\left( \lambda_i^2 - \frac{1}{\lambda_1 \lambda_2} \right) \tag{1}$$

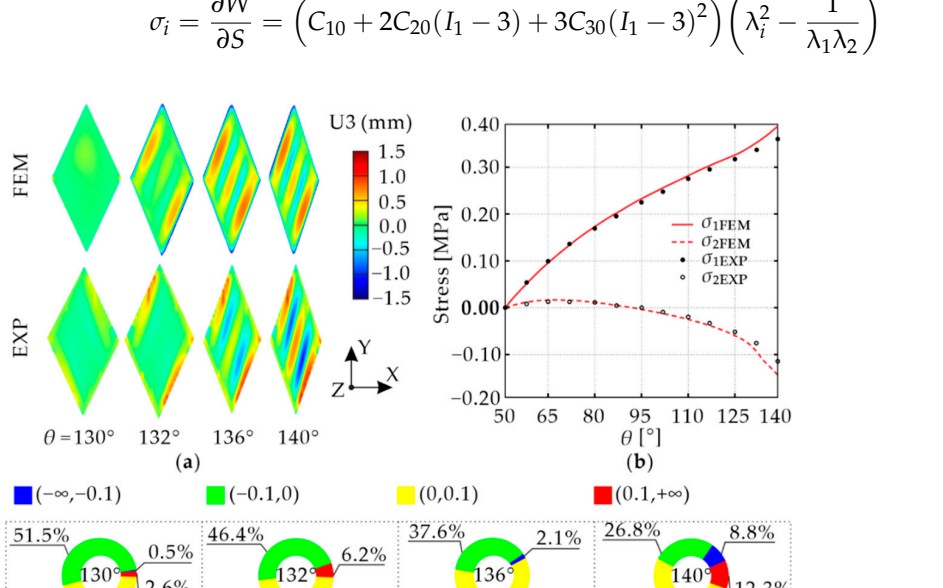

**Figure 6.** Comparison of numerical and experimental results: (**a**) deformation process; (**b**) principal stress versus morphing angle curves; (**c**) pie graphs of the configuration errors.

In the equation, $W = C_{10}(I_1 - 3) + C_{20}(I_1 - 3)^2 + C_{30}(I_1 - 3)^3$ is the strain energy density function, $C_{10}$, $C_{20}$, and $C_{30}$ are material constants listed in Table 2, $I = \lambda_1^2 + \lambda_2^2 + \lambda_3^2$ is the first strain invariant of the Cauchy–Green strain tensor $S$, and $\lambda_i$ ($i = 1, 2, 3$) is the principal extension ratio, which can be defined as:

$$\begin{cases} \lambda_1 = \frac{s}{s_0} = \frac{\sin\frac{\theta}{2}}{\sin\frac{\theta_0}{2}} \\ \lambda_2 = \frac{t}{t_0} = \frac{\cos\frac{\theta}{2}}{\cos\frac{\theta_0}{2}} \\ \lambda_3 = \frac{1}{\lambda_1\lambda_2} = \frac{\sin\frac{\theta_0}{2}\cos\frac{\theta_0}{2}}{\sin\frac{\theta}{2}\cos\frac{\theta}{2}} \end{cases} \tag{2}$$

where $s_0$, $t_0$, and $s$, $t$ are respectively the diagonal lengths before and after deformation, as shown in Figure 7.

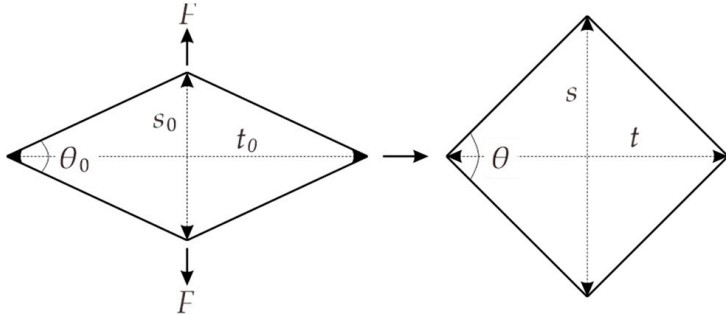

**Figure 7.** Deformation mode of the skin.

Finally, the maximal and minimal principal stress, $\sigma_1$ and $\sigma_2$, at the intersection of the diagonal lines of the module were obtained by substituting the experimental strain values into Equation (1), and are plotted versus the morphing angle $\theta$ as shown in Figure 6b. It should be noted that the strain on the surface of the skin is the same everywhere before wrinkling. In general, a good agreement is obtained. The minor differences at the very large morphing angle are probably caused by fabrication defects of the specimens and the slight difference in boundary conditions. Since the focus of this section is on the critical wrinkling angle of the skins, these minor differences have little influence on the results, so the numerical model is considered acceptable.

## 4. Results

### 4.1. Comparison of Standard and Angled Skin Modules

To understand the deformation mechanism in more detail, numerical models were established, as outlined in this section. The effectiveness of the initial angle is presented here by comparing the standard model $\theta_0 = 90°$ as proposed in reference [31] and the angled one $\theta_0 = 50°$. The numerical deformation processes of the two models are shown in Figure 8a. Moreover, the maximal and minimal principal stresses, $\sigma_1$ and $\sigma_2$, at the intersection point are extracted from the numerical and theoretical models and drawn against the morphing angled $\theta$ in Figures 8b and 8c, respectively.

The deformation of model $\theta_0 = 90°$ is first analyzed to explore the deformation mechanism. Since the deformation below $90°$ is the mirror image of that above $90°$, only the branch above $90°$ is considered. As the morphing angle $\theta$ increases, the compressive stress appears immediately in the skin but no wrinkle shows. This can be attributed to the existence of CFRP rods and KFRP fibers that increase the bending stiffness of the skin and therefore improve stability [31]. When $\theta$ reaches $120.8°$, wrinkles begin to appear, and the magnitude further increases at $\theta = 122°$, consistent with the experimental observation that the model showed no wrinkle at $120°$ but wrinkled at $122°$, as shown in Figure 4a. Moreover, when wrinkles occur at $122°$, it is calculated from the numerical model that the

total strain energy of the CFRP rods is 36.14 mJ, whereas that of the KFRP fibers is 6.83 mJ, implying that CFRP rods play a more important role in delaying wrinkles. Comparing the numerical and theoretical values of $\sigma_1$ and $\sigma_2$ in Figure 8b,c, it can be found that when the skin has a smooth surface, the numerical and theoretical principal stresses match very well. When wrinkles occur, however, the numerical stresses deviate from the theoretical ones. This is because the theoretical values are calculated on the condition that the skin does not buckle, and becomes invalid when it wrinkles.

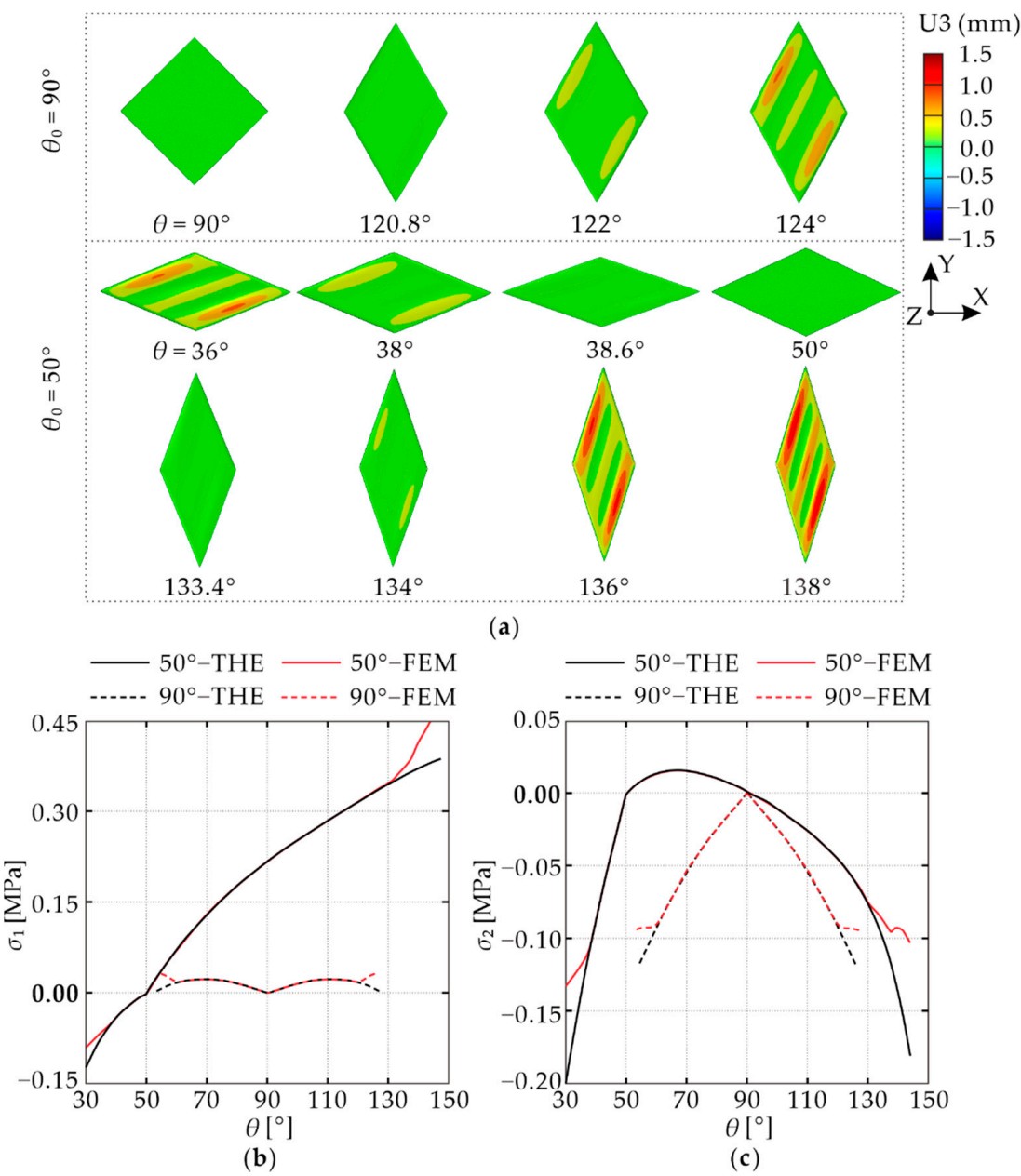

**Figure 8.** (**a**) Deformation process; (**b**) maximal principal stress versus morphing angle curves; (**c**) minimal principal stress versus morphing angle curves.

Subsequently, model $\theta_0 = 50°$ is studied as a comparison. First, consider the branch of increasing $\theta$ from $50°$. The deformation process can be divided into three ranges. First, in the range of $50° \leq \theta \leq 92°$, both $\sigma_1$ and $\sigma_2$ are positive, which indicates that wrinkles will not occur regardless of the presence of reinforcement. This is a significant difference from model $\theta_0 = 90°$, in which compressive stress exists from the beginning of deformation. Second,

when $\theta$ is between 92° and 133.4°, $\sigma_2$ becomes negative and the magnitude gradually increases, but the skin remains smooth, consistent with the experiment result in which specimen M50 was smooth at 130° but wrinkled at 132°, as shown in Figure 4b. At this range, it is mainly the CFRP rods and KFRP fibers that limit the occurrence of wrinkling, and the deformation mechanism is the same as that in model $\theta_0 = 90°$. In addition, model $\theta_0 = 50°$ exhibits a smaller magnitude of compressive stress $\sigma_2$ and a larger magnitude of tensile stress $\sigma_1$ than those of model $\theta_0 = 90°$ throughout this range due to the different initial angle. Previous studies [35] have found that for a plate loaded with a compressive load in one direction, applying a tensile load in the orthogonal direction will increase the critical buckling load and improve the stability of the plate. This explains why model $\theta_0 = 50°$ can be sheared to a larger morphing angle without wrinkling than model $\theta_0 = 90°$. Finally, the skin wrinkles from $\theta = 133.4°$ until the end of loading. When the skin is sheared in the other direction towards $\theta = 38.6°$, both $\sigma_1$ and $\sigma_2$ are compressive stresses, but the smooth configuration is maintained throughout. Comparing the numerical and theoretical principal stresses, the same trend as in the case of model $\theta_0 = 90°$ is obtained, i.e., a good match when the skin is smooth and a noticeable deviation on the occurrence of wrinkles. This implies that we can determine the critical wrinkling angle by comparing the numerical and theoretical principal stresses.

### 4.2. Effect of the Initial Angle

Having demonstrated the effectiveness of the angled skin module design, in this section we investigate the effects of design parameters. First, consider the effect of the initial angle $\theta_0$. The maximal and minimal critical wrinkling angles, $\theta_{max}$ and $\theta_{min}$, of the physical specimens M50–M90, are obtained from experiments and compared in Table 4. As expected, with the reduction in $\theta_0$ from 90° to 50°, $\theta_{max}$ is generally increased whereas $\theta_{min}$ is reduced, leading to a larger wrinkle-free deformation range $\Delta\theta = \theta_{max} - \theta_{min}$. Notice that since the surface of the skin was scanned and examined during the experiment at the interval of 2° until wrinkles occured, the experimental critical wrinkling angles are only approximate values. To obtain the exact critical angles and to explore a wider range of $\theta_0$, seven numerical models listed in Table 4, which have $\theta_0$ from 30° to 90° at an interval of 10°, were built and analyzed. All the other geometric and material parameters are identical to those of the physical specimens. The numerical results are presented in Table 4 and drawn in Figure 9a. Notice that the numerical $\theta_{max}$ and $\theta_{min}$ are determined by comparing the numerical principal stresses at the intersection of diagonals with the corresponding theoretical values calculated from Equation (1). When the numerical values deviate from the theoretical ones, the skin is considered to reach the critical wrinkling angle. It is found that the numerical results match the experimental ones reasonably well with differences less than 2°, once again demonstrating the reliability of the numerical models.

**Table 4.** Comparison of the experimental and numerical results in critical wrinkling angles.

| $\theta_0$ (°) | Experiment | | | FEM | | |
|---|---|---|---|---|---|---|
| | $\theta_{max}$ (°) | $\theta_{min}$ (°) | $\Delta\theta$ (°) | $\theta_{max}$ (°) | $\theta_{min}$ (°) | $\Delta\theta$ (°) |
| 30 | / | / | / | 125.3 | 20.6 | 104.7 |
| 40 | / | / | / | 131.5 | 29.1 | 102.4 |
| 50 | 132.0 | 40.0 | 92.0 | 133.4 | 38.6 | 94.8 |
| 60 | 128.0 | 50.0 | 78.0 | 127.5 | 48.2 | 79.3 |
| 70 | 124.0 | 56.0 | 68.0 | 124.9 | 54.6 | 70.3 |
| 80 | 122.0 | 62.0 | 60.0 | 123.1 | 60.4 | 62.7 |
| 90 | 120.0 | 60.0 | 60.0 | 120.8 | 59.2 | 61.6 |

Moreover, the numerical principal stresses of all the seven models within the wrinkle-free deformation ranges are respectively drawn and compared in Figure 9b,c. It can be seen that the magnitude of the minimum principal stress $\sigma_2$ is reduced as $\theta_0$ becomes smaller at the same $\theta$, and can even turn positive at a certain range when $\theta_0$ is less than

$60°$. In addition, the maximal principal stress $\sigma_1$ increases substantially with the reduction in $\theta_0$. These two factors in combination are accountable for the enlarged $\Delta\theta$. However, as seen in Figure 9c, when $\theta_0$ is too small, it can be seen from the result of $\theta_0 = 30°$ that the minimum principal stress develops quickly with the shear deformation due to the hardening phenomenon of the rubber material under large deformation, which will reduce $\theta_{max}$. In addition, excessive tensile stress will increase the possibility of material failure and lead to a large driving force. Therefore, a too-small $\theta_0$ is also inappropriate.

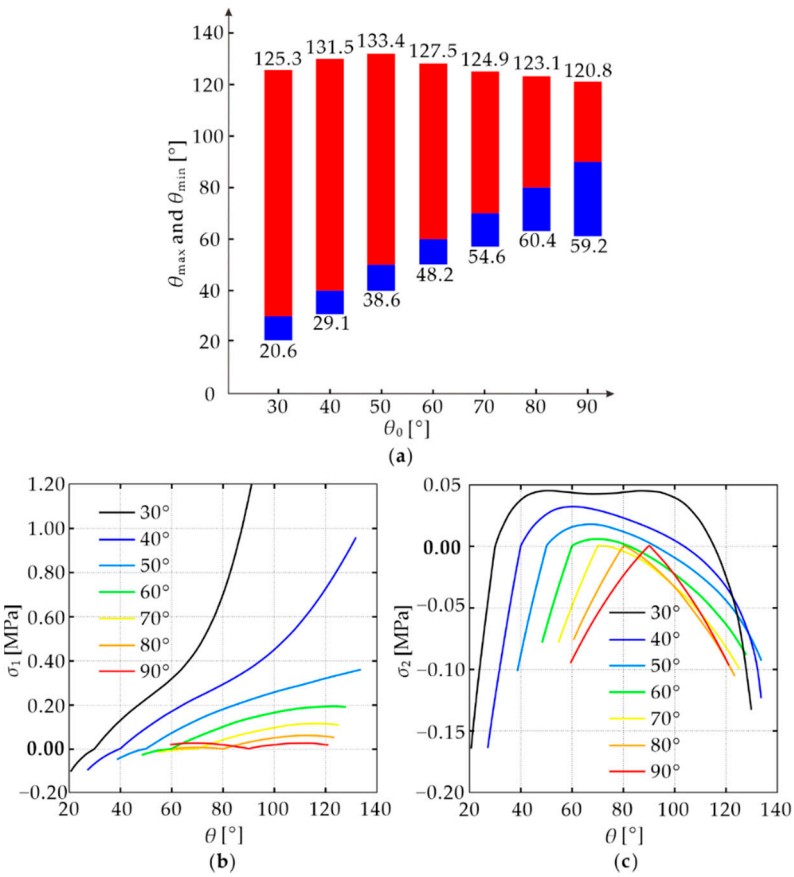

**Figure 9.** (**a**) Maximal and minimal critical wrinkling angles; (**b**) maximal principal stress versus morphing angle curves; (**c**) minimal principal stress versus morphing angle curves.

### 4.3. Effect of the Hardness of Rubber

Subsequently, the effect of the hardness of rubber was analyzed with four numerical models. They have identical geometric and material parameters as those of the physical specimen M50 except for the hardness of the rubber. Rubbers with four different hardness values, 10 HS, 20 HS, 30 HS, and 40 HS, were tested to obtain the engineering stress–strain curves, as shown in Figure 10a, from which the material constants were calculated and listed in Table 5 based on the Yeoh constitutive model. The $\theta_{max}$ and $\theta_{min}$ for each hardness are plotted in Figure 10b. The wrinkle-free deformation range shrinks with the increase in hardness. The reason for the shrinkage can be found by investigating the minimal principal stress $\sigma_2$ at the intersection point of diagonals, which are plotted in Figure 10c. As the rubber hardness increases, the magnitude of $\sigma_2$, which is the main cause of wrinkling, becomes larger at the same morphing angle, thus making it more likely for the skin to wrinkle. Another interesting phenomenon that can be observed from Figure 10c is that within the range of $50° \leq \theta \leq 92°$, $\sigma_2$ is always positive irrespective of the rubber hardness. This is because the sign of $\sigma_2$ is determined by $\lambda_i^2 - 1/\lambda_1^2\lambda_2^2$ according to Equation (1), which is only related to the initial geometry but not the material properties.

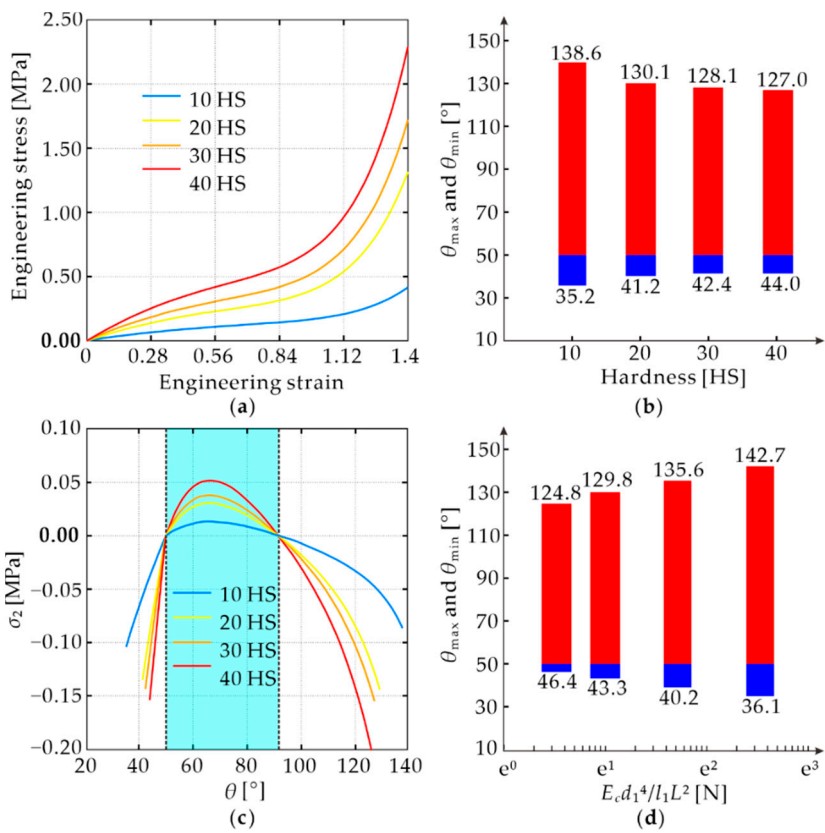

**Figure 10.** (**a**) Material properties of the rubbers with different hardness; (**b**) effects of rubber hardness; (**c**) minimal principal stress at different hardness; (**d**) effects of CFRP reinforcement.

**Table 5.** Material properties of the rubbers with different hardness.

| Hardness/HS | $C_{10}$ | $C_{20}$ | $C_{30}$ | $D$ |
|---|---|---|---|---|
| 10 | 0.04467 | $-0.005278$ | 0.0007991 | 0 |
| 20 | 0.10620 | $-0.015063$ | 0.0029714 | 0 |
| 30 | 0.13025 | $-0.018224$ | 0.0035482 | 0 |
| 40 | 0.17823 | $-0.024546$ | 0.0048051 | 0 |

*4.4. Effect of the CFRP Reinforcement*

It was shown in Section 4.1 that the CFRP reinforcement plays a major role in preventing the skin from wrinkling when compressive stress occurs. Here we investigate its effects through a series of numerical models listed in Table 6, which have various Young's modulus $E_c$, diameter $d_1$, length $L$, and interval $l_1$ of the CFRP rods. All the other geometric and material parameters are identical to those of the physical specimen M50. The results also presented in Table 6 indicate that the $\theta_{max}$ and $\theta_{min}$ of models with the same value of $E_c d_1{}^4 / l_1 L^2$ are also very close, and $\Delta\theta$ increases with this value. The reason is that the wrinkling of the skin is always accompanied by the buckling of the CFRP rods. Along the direction of the CFRP rods placed, the cross-sectional bending stiffness per unit length can be calculated as $E_c I / l_1$, where the area moment of inertia $I = \pi d_1{}^4 / 64$. Since the buckling resistance of the rod is proportional to its bending stiffness and inversely proportional to the square of its length, increasing $E_c d_1{}^4 / l_1 L^2$ would delay the buckling of the CFRP reinforcement and therefore improve the wrinkle-free deformation range of the skin, as seen from the relationship between the critical wrinkling angles and $E_c d_1{}^4 / l_1 L^2$ in Figure 10d. It should be noted that the above analysis is valid only when the interval of the rods is small enough so that there is no local wrinkling between two CFRP rods. In addition, when $L$ is very large, it would be difficult to achieve a large $\Delta\theta$ by adjusting $E_c$, $d_1$, and $l_1$ as

they are limited by the material, skin thickness, and fabrication. Therefore, in the practical application, the length of the rods should be limited.

**Table 6.** Maximal and minimal critical wrinkling angle of the skins with various CFRP reinforcement parameters.

| Group | $E_c$ (GPa) | $d_1$ (mm) | $l_1$ (mm) | $L$ (mm) | $E_c d_1^4/l_1 L^2$ (N) | $\theta_{max}$ (°) | $\theta_{min}$ (°) | $\Delta\theta$ (°) |
|---|---|---|---|---|---|---|---|---|
| A | 675 | 1.5 | 7 | 210 | | 142.7 | 36.0 | 106.7 |
| | 300 | 1.84 | 7 | 210 | 11.07 | 142.4 | 36.2 | 106.2 |
| | 300 | 1.5 | 3.11 | 210 | | 143.0 | 35.7 | 107.3 |
| | 300 | 1.5 | 7 | 140 | | 142.8 | 36.3 | 106.5 |
| B | 533 | 1.5 | 7 | 280 | | 135.9 | 40.3 | 95.6 |
| | 300 | 1.73 | 7 | 280 | 4.92 | 136.0 | 39.8 | 96.2 |
| | 300 | 1.5 | 3.94 | 280 | | 135.1 | 39.3 | 95.8 |
| | 300 | 1.5 | 7 | 210 | | 135.4 | 40.4 | 95.0 |
| C | 168 | 1.5 | 7 | 210 | | 129.8 | 43.5 | 86.8 |
| | 300 | 1.30 | 7 | 210 | 2.77 | 129.6 | 43.2 | 86.4 |
| | 300 | 1.5 | 12.44 | 210 | | 130.1 | 42.9 | 87.2 |
| | 300 | 1.5 | 7 | 280 | | 129.9 | 43.7 | 86.2 |
| D | 108 | 1.5 | 7 | 210 | | 124.8 | 46.5 | 78.3 |
| | 300 | 1.16 | 7 | 210 | 1.77 | 124.9 | 46.2 | 78.7 |
| | 300 | 1.5 | 19.44 | 210 | | 124.5 | 46.7 | 77.8 |
| | 300 | 1.5 | 7 | 350 | | 125.1 | 46.1 | 79.0 |

## 5. Theoretical Analysis

### 5.1. Theoretical Modeling

From the design point of view, a mathematical formula is desired to estimate the critical wrinkling angles of the skin concerning the specified geometry and material. However, the strong geometry and material nonlinearity involved in the deformation process make it very difficult to build a rigorous analytical model. Therefore, a simplified theoretical analysis is performed based on the following assumptions:

(i) The skin is considered as a constructional orthogonal anisotropic plate [36] since the CFRP rods and KFRP fibers are evenly spaced in the rubber. The four edges of the skin are considered to be simply supported. This is close to the realistic boundary condition as the frame provides a limited constraint on the rotation of the skin edge.

(ii) The shear deformation process of the skin is divided into two stages as shown in Figure 11. At stage I, the skin is sheared from the initial angle to 90°. It is known from the previous results that the skin is guaranteed to be wrinkle-free at the end of this stage and is subjected to uniform biaxial in-plane loads and a shear load, denoted as $F_x^{90}$, $F_y^{90}$, and $F_{xy}^{90}$ as shown in Figure 11. Note that the dimension of the three loads is force per unit length along the edges of the plate. At stage II, the skin is further sheared in either direction until it wrinkles. At this stage, we consider the skin as a pre-loaded plate and calculate the critical shear force $F_{xy}^{cr}$ under $F_x^{90}$ and $F_y^{90}$. Then $F_{xy}^{cr} + F_{xy}^{90}$ and $F_{xy}^{cr} - F_{xy}^{90}$ are critical wrinkling forces of the skin in two directions.

(iii) The stresses in the skin are generated only by the rubber since the CFRP rods and KFRP fibers are found from previous results to undergo nearly no deformation before wrinkling.

(iv) The rubber, CFRP rods, and KFPR fibers here are all assumed to be linear elastic materials.

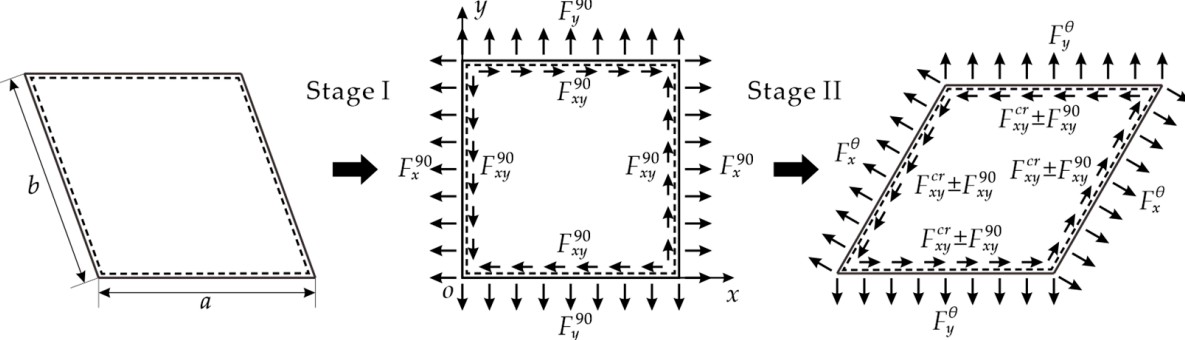

**Figure 11.** Deformation process of the skin.

With the above assumptions, a simplified theoretical model can be derived. First, consider stage I. When an angled model is sheared to the configuration of $\theta$, as shown in Figure 7, the maximal and minimal principal strains, $\varepsilon_1$ and $\varepsilon_2$, can be calculated by the following equation under pure shear deformation:

$$\begin{cases} \varepsilon_1 = \ln\left(\dfrac{\sin\frac{\theta}{2}}{\sin\frac{\theta_0}{2}}\right) \\ \varepsilon_2 = \ln\left(\dfrac{\cos\frac{\theta}{2}}{\cos\frac{\theta_0}{2}}\right) \end{cases} \tag{3}$$

According to assumptions (iii) and (iv), the maximal and minimal principal stresses, $\sigma_1$ and $\sigma_2$, can be obtained as follows [37]:

$$\begin{cases} \sigma_1 = \dfrac{E_r(\varepsilon_1+\mu_r\varepsilon_2)}{1-\mu_r^2} \\ \sigma_2 = \dfrac{E_r(\varepsilon_2+\mu_r\varepsilon_1)}{1-\mu_r^2} \end{cases} \tag{4}$$

in which $E_r$ and $\mu_r$ are equivalent to the Young's modulus and Poisson's ratio of the rubber. Then the stress components $\sigma_x$, $\sigma_y$, and $\tau_{xy}$ can be worked out based on the theory of elasticity [37]:

$$\begin{cases} \sigma_x = \dfrac{\sigma_1+\sigma_2}{2} + \dfrac{\sigma_1-\sigma_2}{2}\cos\theta \\ \sigma_y = \dfrac{\sigma_2+\sigma_1}{2} + \dfrac{\sigma_2-\sigma_1}{2}\cos\theta \\ \tau_{xy} = \dfrac{\sigma_1-\sigma_2}{2}\sin\theta \end{cases} \tag{5}$$

Considering that the total cross-sectional areas of CFRP rods and KFRP fibers account for only a small portion (below 4.3%) of the skin cross-section, the rubber cross-sectional area is regarded to be the same as the cross-sectional area of the skin. Then, when the skin is sheared to $\theta$, the in-plane load, and shear load, $F_x^\theta$, $F_y^\theta$ and $F_{xy}^\theta$, can be expressed as:

$$\begin{cases} F_x^\theta = \sigma_x h \\ F_y^\theta = \sigma_y h \\ F_{xy}^\theta = \tau_{xy} h \end{cases} \tag{6}$$

In which, $h = H\sin\theta_0/\sin\theta$ is the thickness after shearing based on the incompressibility of rubber. By substituting Equations (3)–(5) into Equation (6), $F_x^\theta$, $F_y^\theta$ and $F_{xy}^\theta$ be expressed as:

$$\begin{cases} F_x^\theta = \dfrac{E_r H\sin\theta_0}{2\left(1-\mu_r^2\right)\sin\theta}\left[(1+\mu_r)\ln\left(\dfrac{\tan\frac{\theta}{2}}{\tan\frac{\theta_0}{2}}\right) + (1-\mu_r)\cos\theta\ln\left(\dfrac{\tan\frac{\theta}{2}}{\tan\frac{\theta_0}{2}}\right)\right] \\ F_y^\theta = \dfrac{E_r H\sin\theta_0}{2\left(1-\mu_r^2\right)\sin\theta}\left[(1+\mu_r)\ln\left(\dfrac{\tan\frac{\theta}{2}}{\tan\frac{\theta_0}{2}}\right) - (1-\mu_r)\cos\theta\ln\left(\dfrac{\tan\frac{\theta}{2}}{\tan\frac{\theta_0}{2}}\right)\right] \\ F_{xy}^\theta = \dfrac{E_r H\sin\theta_0}{2(1+\mu_r)}\ln\left(\dfrac{\tan\frac{\theta}{2}}{\tan\frac{\theta_0}{2}}\right) \end{cases} \tag{7}$$

In addition, $F_x^{90}$, $F_y^{90}$, and $F_{xy}^{90}$ at the end of the stage I can also be obtained by substituting $\theta = 90°$ into Equation (7).

At stage II, the critical wrinkling force of the skin is evaluated using the energy method. Since the skin is simply supported at the four edges, the double sinusoidal series [38] satisfying the boundary conditions are chosen as its displacement function $w$, in which $A_{mn}$ is the reciprocally independent coefficient, and $m$ and $n$ are the numbers of terms in the series.

$$w = \sum_{m=1}^{\infty} \sum_{n=1}^{\infty} A_{mn} \sin \frac{m\pi x}{a} \sin \frac{n\pi y}{b} \tag{8}$$

The strain energy $U$ of the skin and the potential energy $V$ of the in-plane loads $F_x^{90}$, $F_y^{90}$, and $F_{xy}^{cr}$, can be expressed as follows [38]:

$$U = \frac{1}{2} \iint \left[ D_1 \left( \frac{\partial^2 w}{\partial x^2} \right)^2 + D_2 \left( \frac{\partial^2 w}{\partial y^2} \right)^2 + 2D_3 \frac{\partial^2 w}{\partial x^2} \frac{\partial^2 w}{\partial y^2} + 4D_k \left( \frac{\partial^2 w}{\partial x \partial y} \right)^2 \right] dxdy \tag{9}$$

$$V = \frac{1}{2} \iint \left[ F_x^{90} \left( \frac{\partial w}{\partial x} \right)^2 + F_y^{90} \left( \frac{\partial w}{\partial y} \right)^2 + 2F_{xy}^{cr} \frac{\partial w}{\partial x} \frac{\partial w}{\partial y} \right] dxdy \tag{10}$$

Substituting Equation (8) into Equations (9) and (10), $U$ and $V$ can be written as:

$$U = \frac{\pi^4}{8} \sum_{m=1}^{\infty} \sum_{n=1}^{\infty} A_{mn}^2 \left[ \frac{m^4 b}{a^3} D_1 + \frac{n^4 a}{b^3} D_2 + \frac{2m^2 n^2}{ab} (D_3 + 2D_k) \right] \tag{11}$$

$$V = \sum_{m=1}^{\infty} \sum_{n=1}^{\infty} A_{mn} \left[ A_{mn} \frac{\pi^2}{8} \left( \frac{m^2 b}{a} F_x^{90} + \frac{n^2 a}{b} F_y^{90} \right) + 8F_{xy}^{cr} \sum_{i=1}^{\infty} \sum_{j=1}^{\infty} A_{ij} \frac{mnij}{(m^2 - i^2)(n^2 - j^2)} \right] \tag{12}$$

where $m$, $n$, $i$, and $j$ satisfy $m \pm i =$ odd and $n \pm j =$ odd. In addition, the bending stiffness $D_1$, $D_2$, $D_3$, and shear stiffness $D_k$ in the equation can be calculated based on assumption (i) [36]:

$$\begin{cases} D_1 = \frac{E_r h^3}{12\left(1 - \mu_r^2\right)} + \frac{E_c I_c}{l_1} \\ D_2 = \frac{E_r h^3}{12\left(1 - \mu_r^2\right)} + \frac{E_k I_k}{l_2} \\ D_3 = \frac{E_r h^3}{12\left(1 - \mu_r^2\right)} \\ D_k = \frac{E_r}{2(1 + \mu_r)} \frac{h^3}{12} \end{cases} \tag{13}$$

in which $E_c$, $I_c$, and $l_1$ are the Young's modulus, the area moment of inertia, and interval of CFRP rods, and $E_k$, $I_k$, and $l_2$ are the Young's modulus, the area moment of inertia, and interval of the KFRP fibers.

Then, the total potential energy of the skin is:

$$\prod = U + V \tag{14}$$

When the skin is in a critical state, according to the principle of minimum potential energy, the following equation is applied:

$$\frac{\partial \prod}{\partial A_{mn}} = 0 \tag{15}$$

From which a set of homogeneous linear algebraic equations expressed in the matrix form $[\mathbf{K}][\mathbf{A}] = [0]$ can be derived. The condition for the existence of non-zero solutions for $A_{mn}$ is $|\mathbf{K}| = 0$. Then, $F_{xy}^{cr}$ can be readily determined. From this, the critical wrinkling angle corresponding to $F_{xy}^{cr} + F_{xy}^{90}$ and $F_{xy}^{cr} - F_{xy}^{90}$ can be solved by Equation (7).

### 5.2. Comparison with Numerical Results

Based on Equation (7), the $\theta_{max}$ and $\theta_{min}$ of the numerical models in Sections 4.2–4.4 were calculated and compared with the results in Tables 7–9, respectively. In the theoretical calculation, the equivalent Young's modulus of the rubber $E_r$ is estimated based on the same strain energy throughout the deformation process.

$$\frac{1}{2}\int_V (\sigma_1\varepsilon_1 + \sigma_2\varepsilon_2)dV = \int_V WdV \tag{16}$$

**Table 7.** Comparison of the theoretical and numerical results for different $\theta_0$.

| $\theta_0$ (°) | Theory | | FEM | | Error | |
|---|---|---|---|---|---|---|
| | $\theta_{max}$ (°) | $\theta_{min}$ (°) | $\theta_{max}$ (°) | $\theta_{min}$ (°) | $\theta_{max}$ | $\theta_{min}$ |
| 50 | 146.8 | 33.2 | 133.4 | 38.6 | 10.0% | 13.9% |
| 60 | 133.6 | 46.6 | 127.5 | 48.2 | 4.8% | 3.3% |
| 70 | 121.7 | 58.3 | 124.9 | 54.6 | 2.6% | 6.8% |
| 80 | 112.9 | 67.1 | 123.1 | 60.4 | 8.3% | 11.1% |
| 90 | 109.4 | 70.6 | 120.8 | 59.2 | 9.4% | 9.4% |

**Table 8.** Comparison of the theoretical and numerical results for different hardness.

| Hardness (HS) | Theory | | FEM | | Error | |
|---|---|---|---|---|---|---|
| | $\theta_{max}$ (°) | $\theta_{min}$ (°) | $\theta_{max}$ (°) | $\theta_{min}$ (°) | $\theta_{max}$ | $\theta_{min}$ |
| 10 | 151.2 | 28.7 | 138.6 | 35.2 | 9.1% | 18.5% |
| 20 | 133.7 | 46.0 | 130.1 | 41.2 | 2.8% | 11.6% |
| 30 | 132.1 | 47.9 | 128.1 | 42.4 | 3.1% | 12.9% |
| 40 | 129.5 | 49.5 | 127.0 | 44.0 | 2.0% | 12.5% |

**Table 9.** Comparison of the theoretical and numerical results for different CFRP reinforcement.

| $E_c d_1{}^4/l_1 L^2$ (N) | Theory | | FEM | | Error | |
|---|---|---|---|---|---|---|
| | $\theta_{max}$ (°) | $\theta_{min}$ (°) | $\theta_{max}$ (°) | $\theta_{min}$ (°) | $\theta_{max}$ | $\theta_{min}$ |
| 11.069 | 151.1 | 28.9 | 142.7 | 36.1 | 5.9% | 19.9% |
| 4.919 | 139.5 | 40.5 | 135.6 | 40.2 | 2.9% | 0.1% |
| 2.767 | 132.8 | 47.1 | 129.8 | 43.3 | 2.3% | 8.8% |
| 1.771 | 128.8 | 48.7 | 124.8 | 46.4 | 3.2% | 4.9% |

By substituting Equations (3) and (4) into Equation (15), $E_r$ can be obtained as follows:

$$\frac{1}{2}\int_V \left[\frac{E_r}{1-\mu_r^2}\left(\varepsilon_1^2 + 2\mu_r\varepsilon_1\varepsilon_2 + \varepsilon_2^2\right)\right]dV = \int_V WdV \tag{17}$$

where $\mu_r$ is taken as 0.5. To cover the entire range of deformation and reduce the error caused by material nonlinearity, the deformation range is taken as $25° \leq \theta \leq 145°$ when calculating $E_r$.

Overall, a reasonable agreement between the theoretical and numerical results is obtained both qualitatively and quantitatively, with the largest error below 20%. The errors mainly originate from two aspects. First, a linear elastic constitutive model is adopted for the rubber in the theoretical analysis, whereas the real stress versus strain relationship is highly nonlinear with a strong hardening effect at large deformation. Second, the critical shear force is calculated based on the configuration of $\theta = 90°$, whereas the configurations at $\theta_{max}$ and $\theta_{min}$ are far away from it, which is equivalent to ignoring the geometric nonlinearity. Moreover, the predictions of $\theta_{max}$ are more accurate than those of $\theta_{min}$. A

possible explanation for this phenomenon is that the deformation range when calculating $E_r$ is much larger than in $\theta_{\min}$, leading to an overestimated $E_r$ at this time.

## 6. Conclusions

In this paper, the pure shear deformation mechanism and critical wrinkling angles of a type of angled composite skin module with rubber matrix and CFRP/KFRP reinforcement were studied experimentally, numerically, and theoretically. The study found that the wrinkle-free deformation range of the angle skin module is increased through two deformation mechanisms. First, when the initial angle is small, the structure has a certain deformation range where both the maximal and minimal principal stresses are positive, which indicates that wrinkles will not occur regardless of the presence of reinforcement. Second, when compressive principal stress occurs, the large tensile principal stress generated during shear tends to delay the onset of wrinkling. It was found from the experiment that by selecting an initial angle of $50°$, the angled skin module can withstand a $92°$ pure shear deformation without wrinkling, which is more than 53% larger than the existing designs. A parameter analysis based on numerical simulation was also carried out to investigate the effects of geometric and material parameters on the critical wrinkling angles. The results show that a small initial angle, low rubber hardness, and large bending stiffness provided by the CFRP rods can achieve a large wrinkle-free deformation range. Finally, a simplified theoretical model based on the energy method was developed to estimate the critical wrinkling angles of the skin module, and a reasonable agreement with the experimental and numerical results was achieved. In the future, a more accurate theoretical model considering geometric and material nonlinearities will be pursued to estimate the critical wrinkling angle of the composite skin. Other critical issues, such as cyclic fatigue and load-bearing capacity, should also receive attention.

**Author Contributions:** Conceptualization, J.Y. and J.M.; methodology, J.Y. and J.M.; software, J.Y.; validation, J.M.; formal analysis, J.Y. and J.M.; investigation, J.Y. and J.M.; resources, J.M.; data curation, J.Y.; writing—original draft preparation, J.Y. and J.M.; writing—review and editing, J.M.; visualization, J.Y. and J.M.; supervision, J.M.; project administration, J.M.; funding acquisition, J.M. All authors have read and agreed to the published version of the manuscript.

**Funding:** This research was funded by the National Natural Science Foundation of China, grant numbers 52192631 and 51721003.

**Institutional Review Board Statement:** Not applicable.

**Informed Consent Statement:** Not applicable.

**Data Availability Statement:** The data presented in this study are available in this article.

**Acknowledgments:** We also thank Zhenhao Jia for the good assistance in CAE.

**Conflicts of Interest:** The authors declare no conflict of interest.

## Nomenclature

| | |
|---|---|
| $L$ | Length of the skin module |
| $T$ | Width of the skin module |
| $H$ | Thickness of the skin module |
| $l$ | Distance between the CFRP rods and the skin top surface |
| $l_1, l_2$ | Intervals of CFRP rods and KFRP fibers |
| $d_1, d_2$ | Diameter of CFRP rods and KFRP fibers |
| $\theta, \theta_0$ | Morphing angle and initial angle of the skin module |
| $\theta_{\max}, \theta_{\min}$ | Maximal and minimal critical wrinkling angles |
| $\Delta\theta$ | Deformation range without wrinkling |
| $\rho$ | Density |

| | |
|---|---|
| $E_r, E_c, E_k$ | Young's modulus of rubber, CFRP rods, and KFRP fibers |
| $\mu_r, \mu_c, \mu_k$ | Poisson's ratio of rubber, CFRP rods, and KFRP fibers |
| $C_{10}, C_{20}, C_{30}$ | Material constants of rubber |
| $\sigma_i$ ($i = 1, 2$) | Maximal and minimal principal stress |
| $W$ | Strain energy density function |
| $S$ | Cauchy-Green strain tensor |
| $I_1$ | First strain invariant of Cauchy-Green strain tensor |
| $\lambda_i$ ($i = 1, 2, 3$) | Principal extension ratio |
| $s_0, t_0$ | Diagonal lengths of the skin module before deformation |
| $s, t$ | Diagonal lengths of the skin module after deformation |
| $F_x^\theta, F_y^\theta, F_{xy}^\theta$ | In-plane loads, and shear load when the skin is sheared to θ |
| $h$ | Thickness of the skin module after deformation |
| $\varepsilon_1$ ($i = 1, 2$) | Maximal and minimal principal strain |
| $\sigma_x, \sigma_y, \tau_{xy}$ | Stress components |
| $w$ | Displacement function |
| $a, b$ | Edge length of the skin module |
| $U$ | Strain energy of the skin module |
| $V$ | Potential energy of the skin module |
| $D_i$ ($i = 1, 2, 3, k$) | Bending stiffness and shear stiffness |
| $A_{mn}$ | Coefficient of displacement function |
| $I_c, I_k$ | Area moment of inertia of the CFRP rods and KFRP fibers |
| $\Pi$ | Total potential energy |

## Appendix A

The effect of biased CFRP rods' placement on the surface roughness of the skin module is discussed in this section. We conducted experiments on two physical specimens with identical $\theta_0 = 60°$. In one specimen, the CFRP rods were placed in the middle along with the skin thickness, and the distance between the rods and the top surface of the skin was $l = 3$ mm. For the other, the placement of the rods was biased, leading to $l = 4.5$ mm. We measured the exact deformed configuration of the top surfaces of the two specimens when they were sheared to 50°, and the U3 contour maps are shown in Figure A1. For the non-biased specimen in Figure A1a, the maximal and minimal displacements in the z-direction are respectively U3$_{max}$ = 0.17 mm and U3 $_{min}$ = 0.03 mm, and thus the surface roughness U3$_{max}$ − U3$_{min}$ = 0.14 mm. For the biased specimen in Figure A1b, the roughness is reduced to 0.05 $_{mm}$, indicating that placing the rods away from the skin top surface can improve its roughness. However, considering that a very large bias will cause the rods to be too close to the rubber bottom surface, which is not conducive to manufacturing, $l = 4.5$ mm was used in the other experiments in this paper.

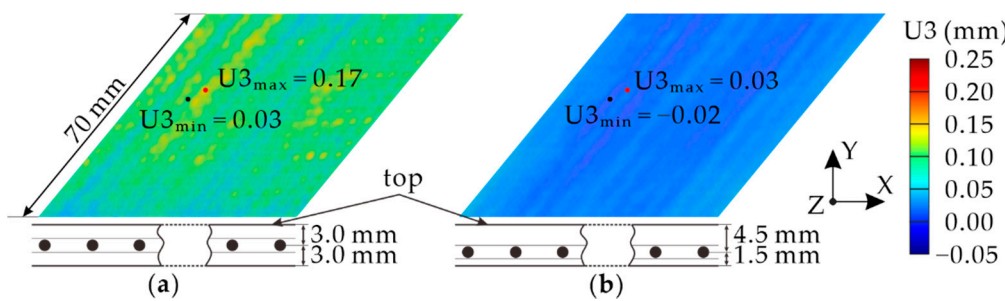

**Figure A1.** Configuration of the skin top surface with $l$: (**a**) $l = 3$ mm; (**b**) $l = 4.5$ mm.

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
