# Peer review of "Design and Shear Analysis of an Angled Morphing Wing Skin Module"

_applsci, doi:10.3390/app12063092_

Round 1

Reviewer 1 Report

The paper proposed and discuss a dedicated technology to design a dedicated material suitable to morphing applications where the level of in plane deformations is very high. I have to admit that I am not a real expert of materials science, however I know by direct experience the problems related to the skin every time we need to design a morphing device. Indeed, any morphing device aiming at keeping the smooth surface must fight against skin. The paper is well written, it is easy to read it and most important is very complete, combining preliminary design, numerical analysis and a very interesting experimental campaign.

My evaluation about the english quality is not very helpful since I am not english mother tongue.

According to me, the paper could be publish as is without significant  modifications.

Reviewer 2 Report

The authors have proposed an angled morphing wing skin module based on a silicon rubber matrix reinforced by carbon-fibre-reinforced polymer rods, which takes advantage of the tensile stress generated during shear to prevent it from wrinkling under large shear deformation.

Detail work on both numerical and experimental have been established to strengthen the defined theoretical model.  Even though the theoretical model can be improved further, the manuscript findings significantly contribute to developing the morphing wings development.

The manuscript is well-written, and I could not appreciate more the authors' effort to ensure all information is explained in depth which enables other researchers to further expand and study/replicate.

Reviewer 3 Report

line 7 - 9 - authors define "improve aerodynamic performance"- this is a broad term, I recommend specifying more precisely

line 27 - authors define the concept "flight conditions" - this is a very inaccurate statement regarding the idea of the article 

line 32 - "flight process" - this is a very inaccurate statement regarding the idea of the article 

the authors use phrases "morphing skin", morphing wing", "morphing aircraft" -  it follows from the references used but it is taken out of context and it is necessary to unify or explain it by definitions

Many mathematical expressions are used in the paper, so I recommend creating the nomenclature (list of symbols or abbreviations)
